# Feature-Enhanced CenterNet for Small Object Detection in Remote Sensing Images

**Tianjun Shi** [1], **Jinnan Gong** [1], **Jianming Hu** [1,*], **Xiyang Zhi** [1], **Wei Zhang** [1], **Yin Zhang** [2], **Pengfei Zhang** [1] **and Guangzheng Bao** [1]

1. Research Center for Space Optical Engineering, Harbin Institute of Technology, Harbin 150001, China
2. Key Laboratory of Space Photoelectric Detection and Perception (Nanjing University of Aeronautics and Astronautics), Ministry of Industry and Information Technology, Nanjing 211106, China
* Correspondence: hujianming@hit.edu.cn; Tel.: +86-0451-86414883

**Abstract:** Compared with anchor-based detectors, anchor-free detectors have the advantage of flexibility and a lower calculation complexity. However, in complex remote sensing scenes, the limited geometric size, weak features of objects, and widely distributed environmental elements similar to the characteristics of objects make small object detection a challenging task. To solve these issues, we propose an anchor-free detector named FE-CenterNet, which can accurately detect small objects such as vehicles in complicated remote sensing scenes. First, we designed a feature enhancement module (FEM) composed of a feature aggregation structure (FAS) and an attention generation structure (AGS). This module contributes to suppressing the interference of false alarms in the scene by mining multiscale contextual information and combining a coordinate attention mechanism, thus improving the perception of small objects. Meanwhile, to meet the high positioning accuracy requirements of small objects, we proposed a new loss function without extra calculation and time cost during the inference process. Finally, to verify the algorithm performance and provide a foundation for subsequent research, we established a dim and small vehicle dataset (DSVD) containing various objects and complex scenes. The experiment results demonstrate that the proposed method performs better than mainstream object detectors. Specifically, the average precision (AP) metric of our method is 7.2% higher than that of the original CenterNet with only a decrease of 1.3 FPS.

**Keywords:** small object detection; remote sensing; CenterNet framework; multiscale information; attention mechanism

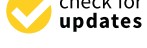



## 1. Introduction

Automatic object detection in remote sensing images has attracted increasing attention in commercial and military fields, which can be widely applied in aerial reconnaissance, traffic monitoring, and area surveillance applications. However, due to the limitations of the resolution and quality of remote sensing images, most objects of interest such as vehicles [1–4] show the following characteristics: small in size, dim in features, and low in contrast [5,6]. In addition, the unique remote sensing imaging system leads to the complexity of the scene and the variability of the target orientation, which brings great difficulties to the detection task. Therefore, it is of great significance to study an effective detection method for small objects in remote sensing images. In this article, we define an object with an area of less than $32 \times 32$ pixels as a small object [7] and concentrate mainly on small vehicle detection in remote sensing images.

A convolution neural network (CNN) [8] can realize end-to-end detection by adaptively learning representative features without handcrafted features. The typical detection networks can roughly be divided into two categories: anchor-based and anchor-free detectors. The anchor-based detectors, such as Faster-RCNN [9] and YOLO [10], require fine-tuned anchor parameters according to the aspect ratio of the objects in the dataset to

achieve promising performances. However, the aspect ratio of objects in different remote sensing scenes appears so diversified, which makes it hard and consuming to adjust the parameters of the anchors [11]. Without the concern of anchor selection, the anchor-free detectors are independent of the hyperparameters of the anchors, reducing the calculation complexity of the algorithm. In addition, the anchor-free detectors detect objects from high-resolution feature maps relying on key points, which are prone to capture objects on a small scale.

CenterNet [12], as a typical representative of anchor-free detectors, directly predicts the center point of the object through the extracted feature map. Compared to other methods, the concise object detection framework of CenterNet gives it the potential to achieve a balance between detection accuracy and speed. In addition, CenterNet uses a high-resolution feature downsampled four times from the input image to predict, which has the advantage of achieving the desired detection performance for small and dense objects. Nevertheless, due to the diversity and complexity of the scenes and the monotonic appearance of the small objects, it is difficult to extract robust features for adequate representations as the performance of CenterNet largely relies on the acquired feature map, which limits its application performance in complex remote sensing scenes to a certain extent.

In this paper, we propose the feature-enhanced CenterNet (FE-CenterNet) by designing a feature-enhanced module (FEM) to help the network reinforce the practical feature while suppressing unnecessary details. At the same time, we adopted a new loss function in the CenterNet framework to ensure the positioning accuracy of small objects. All the above improvements were implemented without many additional parameters and computation costs. In addition, to evaluate the performance of small object detectors in remote sensing images, we constructed a dim and small vehicle dataset (DSVD) composed of various objects and complex scenes. The experiments prove that FE-CenterNet has a significant advantage in small object detection and achieves state-of-art performance on the DSVD.

The main contributions of our work are listed as follows:

- An anchor-free detector, which has excellent performance for small object detection in complex remote sensing scenes.
- A feature-enhanced module, which largely contributes to improving the ability of feature extraction and representation of small objects by digging the multiscale feature and integrating the attention mechanism.
- An established small and dim vehicle dataset, which helps to assess the performance of detection algorithms for small objects.

The remainder of this paper is organized as follows. After introducing the related works of small object detection in Section 2, we elaborate on the proposed FE-CenterNet architecture in Section 3. In Section 4, we briefly introduce the constructed dim and small dataset and describe the experiments conducted to compare the performance of the proposed method and that of typical methods. Finally, we summarize and conclude in Section 4.

## 2. Related Works

With the rapid development of deep learning techniques, remote sensing object detection based on convolutional neural networks (CNN) attracts progressive attention. As we know, mainstream methods are divided into anchor-based and anchor-free frameworks. In this section, we introduce the main development trends and analyze the existing problems of the two categories. On this basis, we illustrate the reason for choosing the anchor-free framework and the solution of the proposed method for small object detection in remote sensing images.

### 2.1. Anchor-Based Framework for Object Detection

After 2012, the rise of CNN promoted object detection to a tremendous advancement. By automatically mining significant features, the problems of poor accuracy and redundant computation based on handcrafted feature descriptors can be alleviated. Anchor-based

detectors predict and classify objects on the different candidates generated from various anchors with the hyperparameters of number, size, and aspect ratio. Whether the candidates are produced, the anchor-based frameworks are divided into two-stage and one-stage detectors. The former uses a region proposal network (RPN) [9] to extract the region of interest (ROI) as the first stage and then performs precise bounding box regression and object classification.

R-CNN [13], as the earliest two-stage detector, first uses the selective search method to generate candidates, followed by CNN to extract the features. The issue lies in the requirement of performing feature extraction on all of the candidates individually, which is a repeatedly time-consuming process. To solve the above problem, Fast-RCNN [14] directly extracts features from the overall image and then maps them to the regions of interest. At the same time, in order to reduce the time consumed by the traditional region proposal algorithm, Faster-RCNN [9] introduces the RPN to achieve end-to-end object detection based on deep learning and further simplifies the detection pipeline. Subsequent improvements mainly appear on the basis of the modules of Faster-RCNN. Mask RCNN [15] utilizes RoIAlign instead of RoIPool, which solves the region mismatch problem in the feature map mapping process. Cascade-RCNN [16] determines the positive and negative samples through different IOU (interaction over union) thresholds and cascades multiple networks to optimize the prediction results. For small target detection, Wang et al. [17] combine the region-based fully convolutional networks (R-FCN) and deformable convolution (DCN) to fully utilize the limited information of small vehicles. Zhang et al. [18] utilize the K-means to generate the hyperparameters of anchors. They introduce the modified VGG16 and Soft-NMS into Faster-RCNN to achieve an effective detection performance of small-scale aircraft. The two-stage networks establish a coarse-to-fine object detection method based on the anchor box mechanism and realize the promising detection performance. However, the two-stage procedure significantly increases the computation cost and inference time.

Compared to the two-stage detectors, the one-stage detector treats object detection as a regression problem. It directly regresses the extracted features to obtain the target category probability and position coordinate value. The detectors of YOLO architecture, as the mainstream one-stage object detector, divide the input image into multiple grids of the same size. They classify the suspected object and regress the position based on the bounding boxes centered on the grid. YOLOv2 [10] designs Darknet as the feature extraction network and adds batch normalization (BN) after all of the convolutional layers. YOLOv3 [19] continues to improve the backbone network named Darknet53, which downsamples the feature map through convolution in place of pooling. In order to solve the imbalance of positive and negative samples in the one-stage network, RetinaNet [20] proposes focal loss to adjust the weight of the indistinguishable samples in the loss function. Now, plenty of improved versions [3,21,22] are proposed based on YOLOv3, which apply and combine a large number of advanced detection technology tricks. They gradually achieve an outstanding balance between accuracy and speed. For small target detection, Bashir et al. [23] combine a cyclic generative adversarial network (GAN) to achieve the image super-resolution of the small targets with the YOLO detection architecture. Zhou et al. [24] apply the gamma correction for image preprocessing to brighten the shadow part of the image and propose a feature fusion structure IR-PANet to increase the recognition ability for small targets. Kim et al. [25] propose an efficient channel attention pyramid YOLO (ECAP-YOLO), which adds a detection layer for small object detection.

However, the anchor-based methods, as the earliest significant branch of object detection, largely depend on the number of positive and negative samples and the hyperparameters of anchor boxes in terms of detection performance. For the unique overlooking perspective of remote sensing, the orientation variance of the object makes it necessary to set a large number of anchor boxes with different scales and aspect ratios, which significantly increases the computation complexity of the algorithm. It limits the detection performance and speed in the remote sensing scenes.

*2.2. Anchor-Free Framework for Object Detection*

Anchor-free detectors remove the anchor mechanism and apply key points to generate the candidates. These methods do not need to set hyperparameters for the anchors, reducing the calculation complexity of the algorithm. The anchor-free frameworks based on key points usually detect objects on high-resolution feature maps, so they are inclined to perceive objects with a small scale. Such methods have a higher flexibility and universality for object detection in remote sensing images from a special bird's-eye view. They have the potential to achieve effective detection of dim and small objects in complex scenes.

CornerNet [26] realizes the localization and detection of objects by predicting the diagonal corners. It utilizes the distance between the embedding vectors of corners to match the same target. Based on CornerNet, CentripetalNet [27] introduces a centripetal shift module and cross-star deformable convolution to achieve bounding box prediction with a higher quality. ExtremeNet [28] predicts the central key points with four extreme key points of each category and matches them using violent enumeration. FCOS [29] directly predicts the distance from the four sides of the bounding box to the center. Although the above methods delete the anchor machinima, they all include some complicated postprocessing methods to match the key points. Compared to other methods, CenterNet [12], as a concise object detection framework, achieves a balance between detection accuracy and speed. It predicts the central point heatmap of the object and obtains information such as the length and width through the features around the center. Motivated by the region proposal network (RPN), FII-CenterNet [30] introduces the foreground information to reduce the influence of the complex scenes and concentrate on the objects of interest. In [31], a probabilistic two-stage detector is constructed based on CenterNet with the combination of the object likelihood and a conditional classification score. In [32], CenterNet++ combines the center key points with corner key points which detect an object as a triplet to capture the salient information globally. The above improved methods are inspired mainly by the two-stage detectors, which can improve the detection accuracy to a certain extent. However, the introduction of foreground information or a region proposal destroys the structural simplicity of CenterNet and dramatically increases the computation complexity and inference time. So, we propose FE-CenterNet to achieve a promising detection performance with little increase in the detection time. To enhance the perception ability of small objects, we designed a feature enhancement module integrating the coordinate attention mechanism and multiscale feature extraction. At the same time, a loss function that integrates robust localization information was put forward in the training process, which can improve the regression accuracy of the location without additional computation for inferring.

## 3. Proposed Method

The main architecture of FE-CenterNet is depicted in Figure 1. Similar to CenterNet, the FE-CenterNet utilizes a modified DLA-34 [12] as the backbone to extract features at multiple levels and obtains the feature map downsampled four times. Specifically, we propose the feature-enhanced module (FEM) after the backbone network, improving the representation ability of small object characteristics. This module is composed of feature aggregation structure (FAS) and attention generation structure (AGS), and the detailed explanation of the two structures is provided in Section 3.1. FAS integrates the multiscale feature with the introduction of context information to suppress the false alarms from complex scenes. Moreover, AGS embeds the coordinate relationship into the attention mechanism, strengthening the perception ability of small objects. During the training process, we put forward a new loss function to adapt to the high demand for positioning accuracy, which is elaborated in Section 3.2. The loss function improves detection performance without extra calculation during inference process.

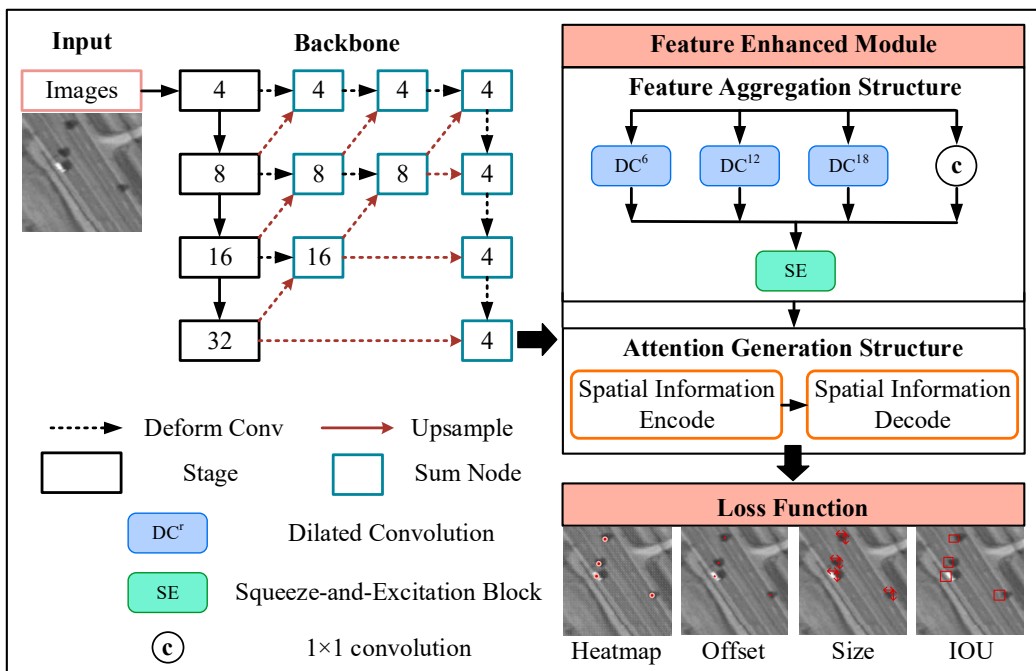

**Figure 1.** The structure of FE-CenterNet.

### 3.1. Feature-Enhanced Module

The detection of small objects in remote sensing images mainly faces two challenges: (1) complicated scenes which contain too many false alarms that interfere with small object detection; and (2) objects are small in scale and weak in characteristics, which makes it tough to capture the practical features. In response to the above problems, we propose a feature enhancement module (FEM), which is comprised the feature aggregation structure (FAS) and the attention generation structure (AGS). This module extracts multiscale features to aggregate contextual information in images and enhances the perception ability of valid features for small-scale objects through the attention mechanism. Through FEM, the feature aggregation and enhancement of the high-resolution feature map, extracted by the backbone extraction network, can effectively improve the detection performance for small objects. The main structure of the feature aggregation enhancement module is shown in Figure 2.

Due to the complexity of remote sensing images, false alarms with similar characteristics to target are prone to appear, largely affecting the detection performance. False alarms have the same characteristics as the target, which is challenging to identify by merely their characteristics. Therefore, it is necessary to introduce global context information and use the semantic information of the scene to suppress false alarms. The ordinary convolution has a fixed receptive field and can only perform feature extraction on a local area with a fixed size. In the feature aggregation structure (FAS), we utilized several dilated convolutions in parallel to gather the multiscale information in the feature map inspired by the ASPP block [33]. The gathered output can be less affected by the complex scenes due to the aggregation of valid semantic information.

Compared to ordinary convolution, dilated convolution gives access to the receptive field of different scales by adjusting the dilation rate. Here, we denote a dilated convolution with the dilation rate of $m$ and the size of a kernel $n \times n$ as $\mathrm{dconv}_{n \times n}^{m}$. For the input feature map $F_{in} \in \mathbb{R}^{c \times h \times w}$, $h$ and $w$ represent the length and width of the feature map, respectively. $c$ is the number of channels. The feature extraction result under a specific receptive field is obtained with the same dimension as the input feature map:

$$F_{n \times n}^{m} = \mathrm{dconv}_{n \times n}^{m}(F_{in}) \tag{1}$$

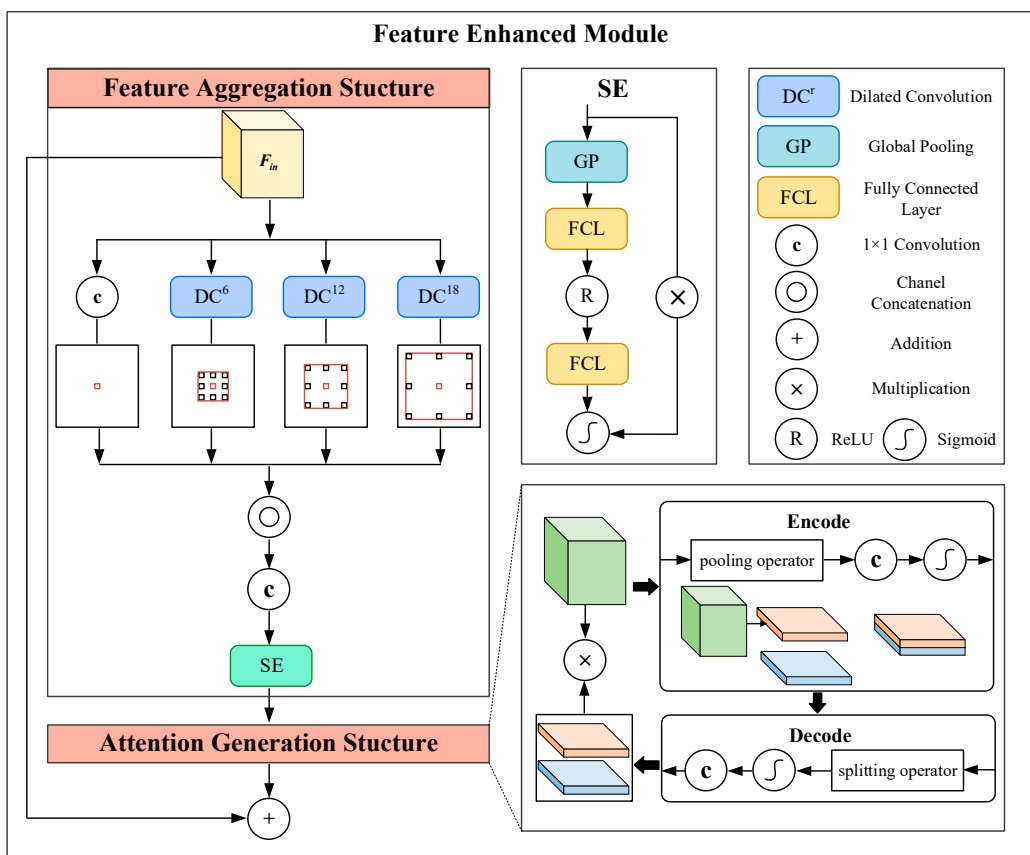

**Figure 2.** The structure of FEM.

We set the dilation rate of 6, 12, and 18 and made the dilated convolution operate on the input feature map to get the aggregated features at different scales. In addition, a $1 \times 1$ convolution was used to keep the feature representation with the exact resolution as the input feature map. Then, concatenation operator and $1 \times 1$ convolution were applied to get the final output. The calculation process is shown below, where $F_{cat}$ is the aggregated result of the multiscale information. Here, $1 \times 1$ convolution helps to make the output of the feature map channels the same as the channels of the input map.

$$F_{cat} = \text{conv}_{1\times 1}\left[\text{dconv}_{3\times 3}^{6}(F_{in}), \text{dconv}_{3\times 3}^{12}(F_{in}), \text{dconv}_{3\times 3}^{18}(F_{in}), \text{conv}_{1\times 1}(F_{in})\right] \quad (2)$$

The extracted multiscale features in different channels make distinct contributions to detecting small objects. Therefore, we added the channel attention mechanism to assign different weights to each channel based on the significance after feature fusion. The channel attention mechanism automatically obtained the importance of each channel through learning, thus strengthening the edge details and semantic information. Inspired by the SE block [34], we obtained the global information of each channel through the spatial pooling operator, yielding a $1 \times 1 \times C$ channel feature vector. The $k$-th channel of the feature vector $v_c$ is formulated as:

$$v_c^k = \frac{1}{h \times w} \sum_{i=1}^{h} \sum_{j=1}^{w} F_{cat}^k(i,j) \quad (3)$$

where $F_{cat}^k(i,j)$ represents the value of $F_{cat}$ in the $i$-th row, the $j$-th column, and the $k$-th channel.

After that, we used the bottleneck layer composed of two fully connected layers. The dimension of the feature vector was first reduced and then restored to the original one. The bottleneck layer can better adapt to the complex correlation between channels and reduce the amount of calculation. The sigmoid function processes the feature vector to obtain the

normalized weight of each channel. Finally, the feature map in each channel was multiplied by the weight factor to obtain the rescaled result. The final output is written as:

$$F_{FAS} = F_{cat} \cdot \text{Sigmoid}(\text{FC}(\text{ReLU}(\text{FC}(v_c)))) \tag{4}$$

where FC represents the fully connected layer, and ReLU and Sigmoid are the nonlinear activation function.

Due to the limited geometric scale, small objects lack texture details. At the same time, the positioning accuracy requirement for small objects is higher than that for large objects. It means that a slight deviation of the center position may lead to inaccurate bounding box regression. Therefore, after using the feature aggregation structure (FAS) to obtain feature maps integrated with the multiscale information, we proposed an attention generation structure (AGS) based on the coordinate attention mechanism. By embedding the coordinate position, this structure strengthened the effective features of the small objects to improve the localization and perception ability.

The attention mechanism helps the network improve the perception of specific detailed features and semantic information by applying different significance to channels and regions. Inspired by the CA [35] block, the AGM consists of spatial information encoding and decoding procedures. The coordinate embedding contributes to mining the spatial dimension information beneficial to localizing small objects.

First, AGS embeds spatial information into the channel relationship through a pair of pooling operators along the dimensions *x* and *y*, respectively. This pooling can preserve the coordination information while obtaining channel description in comparison with global pooling. Due to the embedding of coordinate position, the encoded feature map can capture the spatial information of the interested region, which helps to satisfy the dependency of position information for small object detection. For the feature map $F_{cat} \in \mathbb{R}^{c \times h \times w}$ from the FAS, the pooled vector $v_x \in \mathbb{R}^{c \times h \times 1}$ and $v_y \in \mathbb{R}^{c \times 1 \times w}$, along a single spatial dimension, can be formulated below:

$$v_x^k(i) = \frac{1}{w} \sum_{j=1}^{w} F_{cat}^k(i,j) \tag{5}$$

$$v_y^k(j) = \frac{1}{h} \sum_{i=1}^{h} F_{cat}^k(i,j) \tag{6}$$

where $v_x^k(i)$ is the pooled result along the vertical direction in *k*-th channel and *i*-th position, and $v_y^k(j)$ is the pooled result along the horizontal direction in *k*-th channel and *j*-th position.

For the pooled vectors calculated from Equations (5) and (6), we applied the channel concatenation to obtain the aggregated vector $v_{x,y} \in \mathbb{R}^{c \times 1 \times (h+w)}$. In addition, $1 \times 1$ convolution was utilized to achieve the reduction in channel dimension. This channel compression process facilitates the representation of channel correlations while reducing the number of parameters. The final encoded result $v_{encode} \in \mathbb{R}^{c/r \times 1 \times (h+w)}$ is expressed as:

$$v_{encode} = \text{Sigmoid}(\text{conv}_{1\times 1}(v_{x,y})) \tag{7}$$

where $\text{conv}_{1\times 1}$ denotes the $1 \times 1$ convolutional transformation.

After acquiring the feature vector $v_{encode}$ which encoded the spatial information, the next step was the spatial information decoding and applying the decoded attention weights to the input feature map. The encoded vector $v_{encode}$ was split along vertical and horizontal dimensions to get the single direction encoded vectors $v'_x \in \mathbb{R}^{c/r \times 1 \times h}$ and $v'_y \in \mathbb{R}^{c/r \times 1 \times w}$:

$$v'_x, v'_y = \text{split}(v_{encode}) \tag{8}$$

where split($\cdot$) represents the dimension splitting operator.

For the split vectors, $1 \times 1$ convolutional transformation was used to restore the influence of channel reduction, yielding the same channel dimension as the input feature map. The decoded attention weights along different spatial directions can be written as:

$$w_x = \text{Sigmoid}\big(\text{conv}_{1\times1}\big(v'_x\big)\big) \tag{9}$$

$$w_y = \text{Sigmoid}\big(\text{conv}_{1\times1}\big(v'_y\big)\big) \tag{10}$$

where $w_x$ and $w_y$ are a pair of attention weights embedded in the vertical and horizontal spatial information, respectively. By applying the decoded attention weights, the final output feature map $F_{out} \in \mathbb{R}^{c \times h \times w}$ can be formulated as:

$$F_{AGS}^k(i, j) = w_x^k(k, i) \times w_y^k(k, j) \times F_{FAS}^k(i, j) \tag{11}$$

### 3.2. Loss Function

In order to improve the regression accuracy of bounding boxes for small objects, the original loss of CenterNet was updated with the complete interaction over union (CIOU) [36], which was finally composed of keypoint heatmap, and interaction over union, size, and central offset. The whole function $L_{det}$ is formulated as:

$$L_{det} = L_{heatmap} + \lambda_{size} L_{size} + \lambda_{offset} L_{offset} + \lambda_{ciou} L_{ciou} \tag{12}$$

We set $(1, 0.1, 1)$ to $\big(\lambda_{size}, \lambda_{offset}, \lambda_{ciou}\big)$, which are the hyperparameters to adjust the weight of each part in the loss function.

CenterNet detects objects as points and generates the keypoint heatmap $\hat{P} \in [0, 1]^{W/R \times H/R \times C}$, size prediction $\hat{S} \in \mathbb{R}^{W/R \times H/R \times 2}$, and central offset $\hat{O} \in \mathbb{R}^{W/R \times H/R \times 2}$ before predicting [11], where $W$, $H$, and $C$ represent the width, length, and object categorifies, respectively. $R$ is the downsample stride, and we set it to 4, ensuring sufficiently high-resolution feature maps for small object detection. The keypoint loss is defined as:

$$\begin{cases} L_{heatmap} = \frac{-1}{N} \sum\limits_{xyc} \big(1 - \hat{P}_{xyc}\big)^{\alpha} \log\big(\hat{P}_{xyc}\big) & , P_{xyc} = 1 \\ L_{heatmap} = \frac{-1}{N} \sum\limits_{xyc} \big(1 - \hat{P}_{xyc}\big)^{\beta} \big(\hat{P}_{xyc}\big)^{\alpha} \log\big(1 - \hat{P}_{xyc}\big), otherwise \end{cases} \tag{13}$$

where $P_{xyc}$ is the ground truth heatmap generated by the Gaussian function same as CenterNet. As only those $P_{xyc} = 1$ were viewed as positive samples, it brought about the imbalance between positive and negative samples, which we used the focal loss to alleviate. $\alpha$ and $\beta$ are the hyperparameters in focal loss set to 2 and 4 by default. $N$ is the total number of key points used for normalization

For the $k$-th ground truth bounding box denoted as $\big(x_1^{(k)}, y_1^{(k)}, x_2^{(k)}, y_2^{(k)}\big)$, the length and width is $s_k = \big(w^{(k)}, h^{(k)}\big) = \big(x_2^{(k)} - x_1^{(k)}, y_2^{(k)} - y_1^{(k)}\big)$, while the central position is $p = \big(\big(x_1^{(k)} + x_2^{(k)}\big)/2, \big(y_1^{(k)} + y_2^{(k)}\big)/2\big)$. The size and offset are both trained with L1 loss, which are calculated as

$$L_{size} = \frac{1}{N} \sum_{k=1}^{N} \big| \hat{S}_{pk} - s_k \big| \tag{14}$$

$$L_{offset} = \frac{1}{N} \sum_{p} \big| \hat{O}_{\widetilde{p}} - \big(\frac{p}{R} - \widetilde{p}\big) \big| \tag{15}$$

where $\widetilde{p} = [p/R]$ represents the integer part of position after downsampling by $R$ times.

The original CenterNet loss function independently optimizes the central position and target size, which causes poor positioning accuracy for small objects. Therefore, we introduced the CIOU during the loss function calculation training under the supervision of overlapping between the prediction bounding box and the ground truth bounding

box. CIOU considers the distance, overlap degree, and aspect ratio and comprehensively optimizes the matching degree of the prediction and ground truth bounding boxes. The CIOU is written as:

$$L_{ciou} = 1 - IOU + \frac{\rho^2\left(p_{pred}, p_{gt}\right)}{c^2} + \alpha v \tag{16}$$

where *IOU* is the interaction over union between the prediction bounding box and the ground truth bounding box. $p_{pred}$ and $p_{gt}$ are the center point of prediction and ground truth, respectively. $\rho$ represents the Euclidean distance operator, and $\alpha$ is the weight factor. The aspect ratio similarity is formulated as:

$$v = \frac{4}{\pi^2}\left(\arctan\frac{w^{gt}}{h^{gt}} - \arctan\frac{w}{h}\right)^2 \tag{17}$$

The addition of the CIOU loss can improve the positioning accuracy of CenterNet for small objects and improve the convergence efficiency of the network.

## 4. Experimental Results

### 4.1. Dim and Vehicle Datasets

We built a dim and small vehicle dataset (DSVD) based on the UNICORN 2008 dataset [37] to evaluate the small object detection performance of the proposed algorithm. The UNICORN 2008 source dataset is a kind of wide area motion imagery (WAMI) dataset, which includes 6471 images. Each image has a coverage area of about 5 km × 5 km and an image size of around 10,000 × 10,000 pixels. Based on UNICORN 2008, the constructed small object dataset has the following detection difficulties:

- The vehicles with relatively monotonic appearance are small in size, dim in features, and low in contrast. It is very difficult to obtain a robust characteristic representation for these objects. The local regions of objects are shown in Figure 3.
- The images cover a wide area and various complex scenes, such as parking lots, roads, neighborhoods, etc. In addition, there are plenty of suspected objects in scenes prone to becoming false alarm sources. The complicated scenes are depicted in Figure 4.

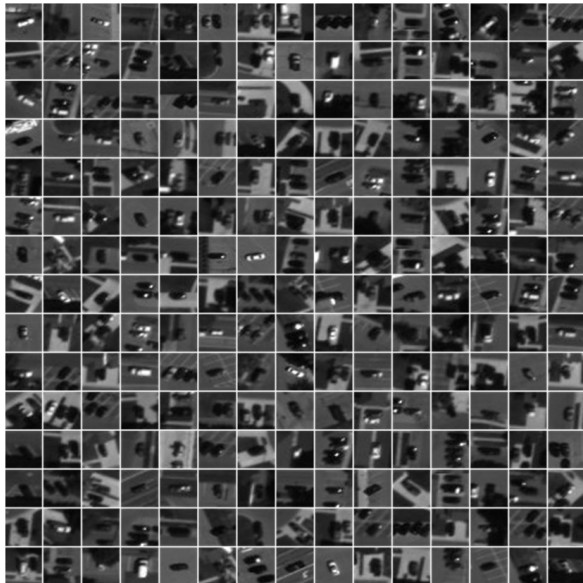

**Figure 3.** The vehicles in UNICORN 2008 dataset.

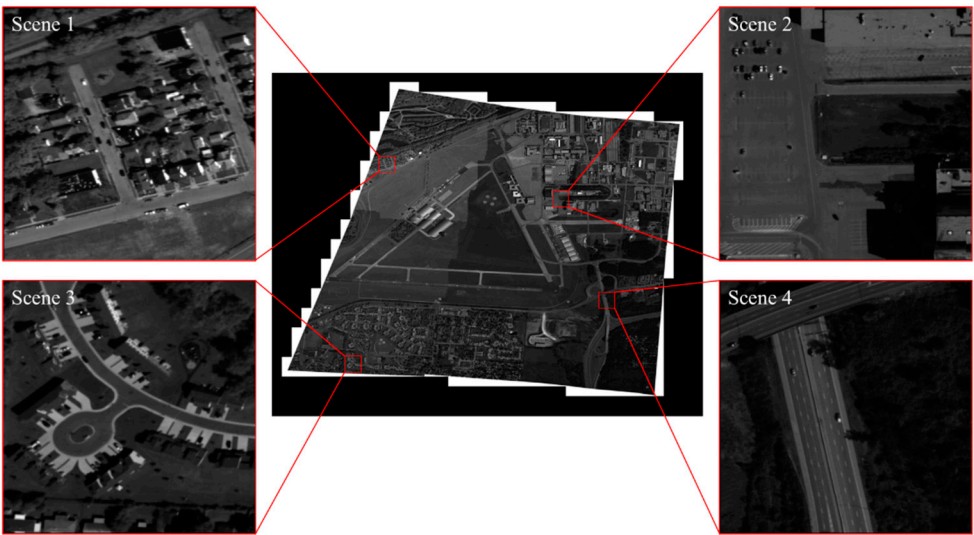

**Figure 4.** The complicated scenes in UNCIRON 2008 dataset.

The aforementioned complexity of scenes and weak characteristics of objects make detecting vehicles in UNICORN 2008 rather challenging. We split the images in UNICORN 2008 into several blocks of 640 × 640 pixels and selected diverse scenes. For the picked images up to 3225, we labeled the vehicles thoroughly using the rectangular bounding box. In total, 2257 images were randomly chosen from the whole images for the network training, and the other 968 images were the test data for the network performance evaluation.

### 4.2. Evaluation Metrics

We applied the precision, recall, F1-score, and AP (average precision) metrics to evaluate the detection performance of the proposed method. The intersection over union (IOU) of the detected bounding boxes and ground truth bounding boxes was set to a threshold of 0.5. Among these metrics, precision and recall can be used to evaluate the detection of missed and false alarms, which are calculated as follows:

$$precision = \frac{TP}{TP + FP} \tag{18}$$

$$recall = \frac{TP}{TP + FN} \tag{19}$$

where $TP$, $FP$, and $FN$ represent the true positive, false positive, and false negative.

The F1 score and $AP$ can more comprehensively evaluate the detection method. F1, the harmonic mean of precision and recall, is written as:

$$\text{F1} = \frac{2}{\frac{1}{precision} + \frac{1}{recall}} \tag{20}$$

$AP$ is defined as the area surrounded by the recall–precision curve, which is formulated as:

$$AP = \int_0^1 precision(recall)d(recall) \tag{21}$$

To evaluate the inference speed of the detection algorithm, we utilized the FPS (frame per second) metric.

### 4.3. Implementation Details and Ablation Analysis

The experiments were all conducted in an Inter Xeon® Silver 4210R CPU and NVIDIA Quadro RTX 4000 GPU with the Pytorch framework. During the training process, the

input resolution was 512 × 512 pixels, and we obtained a 128 × 128 pixels feature map to predict. We trained the model with a batch size of four for 140 epochs. An Adam optimizer was chosen with the learning rate of $8 \times 10^{-5}$, which was reduced by 10 times at 90 and 120 epochs, respectively.

Figures 5 and 6 show some detection results of our method, which can achieve a good performance for small and dim objects. As shown in Figure 5, some objects with limited appearance features are in low contrast, while our method can detect all the objects without missing alarms. For the complicated scenes with plenty of interference similar to the objects in Figure 6, our method can also perform well without false alarms.

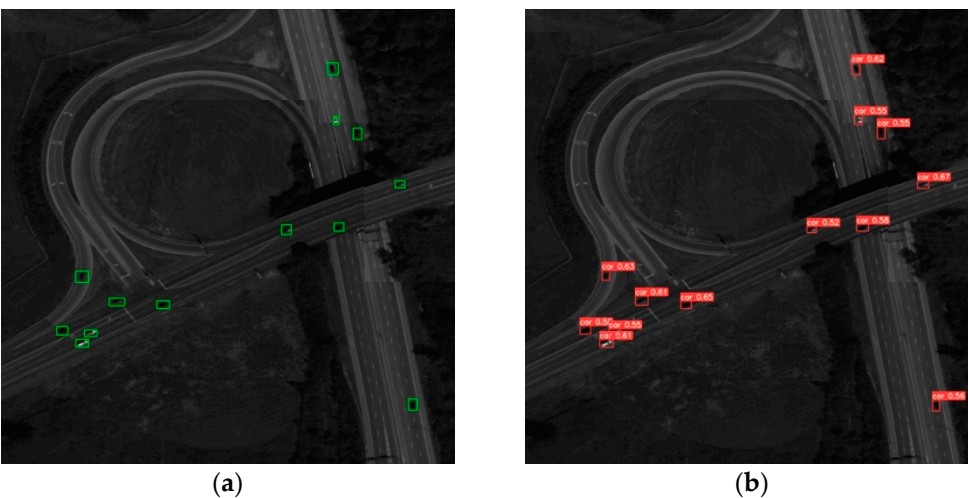

| (**a**) | (**b**) |

**Figure 5.** (**a**) Ground truth and (**b**) detection results. The detection results in scenes with dim targets of the proposed method.

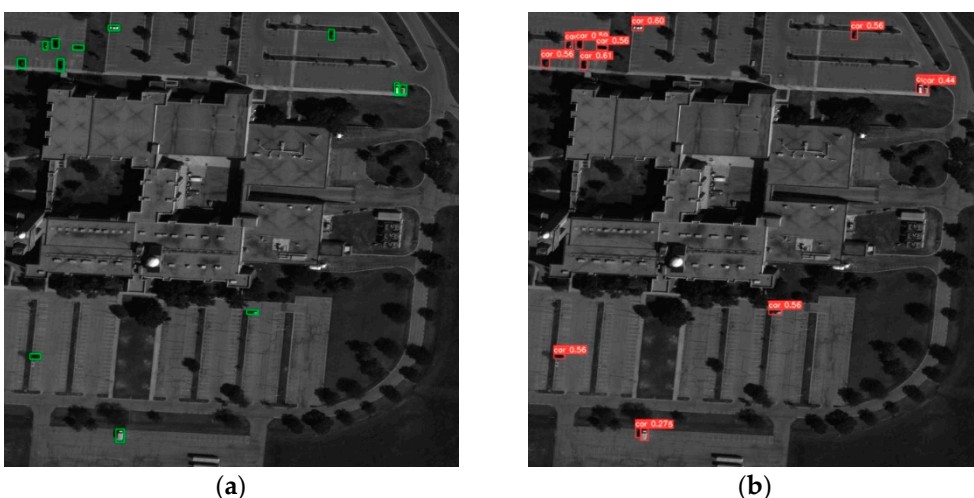

| (**a**) | (**b**) |

**Figure 6.** (**a**) Ground truth and (**b**) detection results. The detection results in complicated scenes of the proposed method.

Based on the constructed DSVD, we conducted ablation experiments on the proposed method to evaluate the performance improvement in small object detection. The same strategy and parameters were applied during the training and evaluation process to ensure fair comparisons. As shown in Table 1, we used the precision, recall, F1-score, and AP to evaluate the detection performance and the FPS to compare the detection speed. Clearly, the proposed method has excellent advantages in terms of the metrics of detection performance compared with CenterNet and hardly affects inference speed. The addition of the FEM and the improvement in the loss function increases the AP metric by 4.3% and 2.7%, with a slight decrease in the FPS metric. Finally, compared with the initial CenterNet,

FE-CenterNet, in this paper, improves the AP by 7.2%, with almost no additional inference time added (the FPS is from 17.9 to 16.6). Meanwhile, to visually illustrate the effect of the FEM, we provide some feature visualization in Figure 7. Before enhancement, there exists plenty of interference which may be highlighted in the feature map. However, they are suppressed a lot after the enhancement of the FEM. In addition, the objects with weak characteristics on the left and right are not clearly shown in the feature map. Through the feature enhancement, these targets are all highlighted.

**Table 1.** Ablation experiments for proposed method.

| Methods | Precision | Recall | F1 Score | AP | FPS |
|---|---|---|---|---|---|
| CenterNet | 79.8% | 74.9% | 77.3% | 70.2% | 17.9 |
| CenterNet + FEM | 82.7% | 78.6% | 80.6% | 74.5% | 16.7 |
| CenterNet + proposed loss function | 80.8% | 77.0% | 78.9% | 72.9% | 17.9 |
| CenterNet + FEM + proposed loss function | 83.5% | 80.8% | 82.1% | 77.4% | 16.6 |

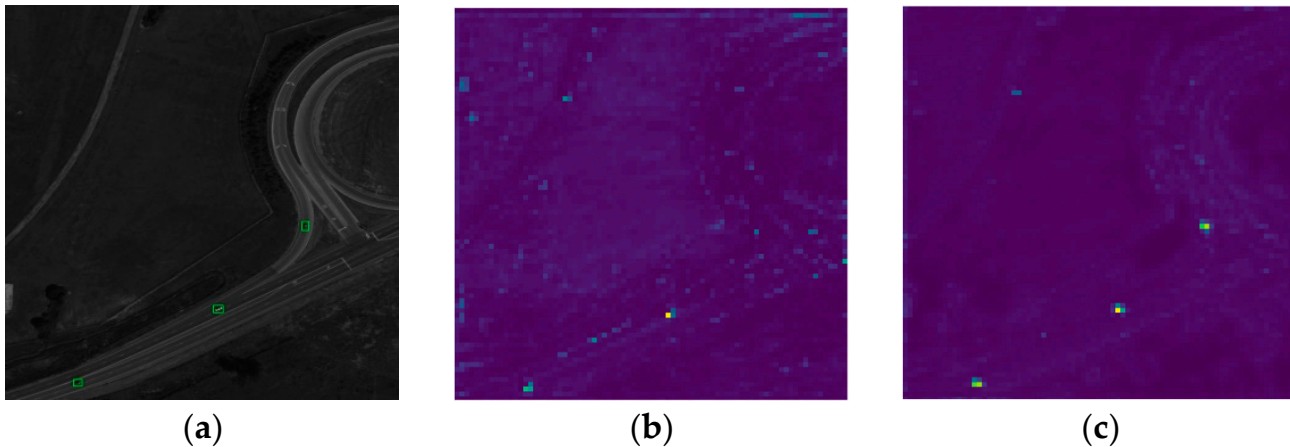

**(a)** **(b)** **(c)**

**Figure 7.** (**a**) Ground truth, (**b**) before enhancement, and (**c**) after enhancement. Visualization results of the FEM.

In order to exhibit the improvement in the detection performance more intuitively, the detection results with CenterNet and the proposed FE-CenterNet are visualized, as depicted in Figure 8. Among them, three typical scenes of roads, parking lots, and communities in the dataset are selected. It can be seen that based on the feature enhancement module and the improved loss function, FE-CenterNet is less susceptible to the interference of false alarm sources and has a stronger perception ability for small objects. In the remote sensing images with complex and changeable scenes, natural landscapes and artificial equipment with similar characteristics to the target are prone to appear, such as shadow blocks (regions 1, 3, 5, 6, 9, and 10), roofs (region 2), and trees (regions 4 and 7). CenterNet cannot make good use of the contextual information, which makes it difficult to distinguish objects from such false alarms. The network proposed in this paper can aggregate the multiscale features through the feature aggregation structure, effectively reducing the mistake of false alarms and improving precision. At the same time, the vehicle objects in the remote sensing image are small in geometric scale and weak in texture and structural characteristics, which is hard to be fully perceived by the network. As shown in region 8 and region 11, CenterNet failed to detect the tiny target. At the same time, FE-CenterNet can achieve a more effective perception of them through the attention generation structure and improve recall.

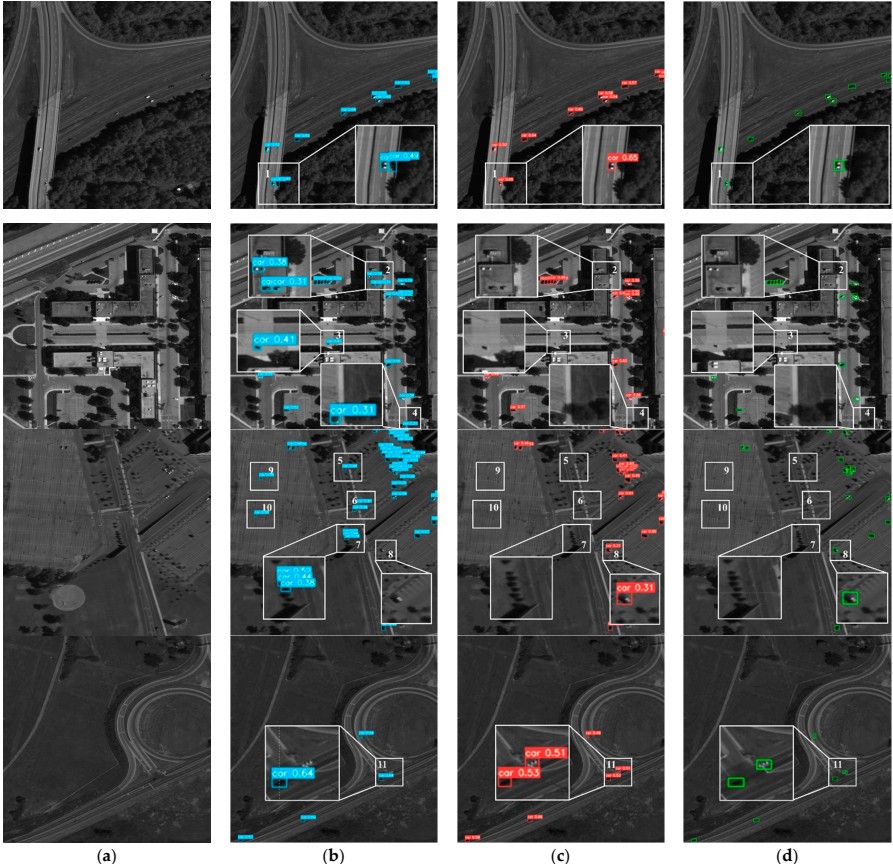

**Figure 8.** (**a**) Input image, (**b**) CenterNet, (**c**) our method, and (**d**) ground truth. Visualization of the detection results in ablation experiments.

### 4.4. Algorithm Performance Comparison

In order to verify the overall performance of our proposed method, we compared it with multiple representative detection algorithms based on the same implementation environment, datasets, and evaluation metrics. Among the selected methods, Cascade-RCNN [15] is a two-stage detector improved from Faster-RCNN, which generally outperforms the one-stage and anchor-free detectors with a higher computational complexity. ImYOLOv3 [3] introduces an attention mechanism to the one-stage YOLOv3, which performs well for remote sensing object detection. YOLOv7 [21] is currently the latest algorithm with the YOLO architecture, which combines plenty of advanced detection tricks. FII-CenterNet [23], also based on the anchor-free CenterNet, improves the detection ability of traffic objects through the foreground region proposal network drawn from the two-stage network.

The evaluation results are shown in Table 2. Among the methods, the coarse-to-fine detection pipeline of Cascade-RCNN and the introduction of foreground information in FII-CenterNet help improve precision but can hardly detect all the objects. The idea of two-stage also affects the inference speed a lot. While imYOLOv3 improves the perception ability of small and dim objects by applying the attention mechanism, the recall is comparatively higher. However, it is prone to interfere with false alarms which are similar to the objects. The YOLOv7 adopts the anchor-free mechanism and has an obvious advantage in speed. The combination of the advanced training and inference strategies also makes it outperform the other comparison algorithms. However, lacking in strategies designed for small objects, it is less effective than our FE-CenterNet. Our method, based on the multiscale feature fusion structure and the attention generation structure, ensures the highest precision, recall, F1, and AP while simultaneously keeping a relatively high detection speed.

**Table 2.** Quantitative results of different methods.

| Methods | Precision | Recall | F1 Score | AP | FPS |
|---|---|---|---|---|---|
| Cascade-RCNN | 81.0% | 69.1% | 74.6% | 75.6% | 7.7 |
| imYOLOv3 | 76.0% | 78.7% | 77.3% | 74.4% | 15.8 |
| YOLOv7 | 78.8% | 79.3% | 79.1% | 76.1% | 20.1 |
| FII-CenterNet | 77.2% | 74.1% | 75.7% | 74.8% | 14.3 |
| FE-CenterNet (ours) | 83.5% | 80.8% | 82.1% | 77.4% | 16.5 |

The above conclusions are intuitively reflected by visualizing the detection results in Figure 9. We display the detection results of imYOLOv3, Cascade-RCNN, FII-CenterNet and the proposed method to compare. We chose the displayed images from various complex scenes such as parking lots, communities, and roads. Meanwhile, the occlusion and low contrast between the objects and scenes caused great difficulty in detection. From region 1 to region 3 where objects are densely distributed, a large number of objects were missed in the detection results of Cascade-RCNN and FII-CenterNet, while imYOLOv3 and the proposed method can detect more objects in such scenes. For objects with occlusion in region 4 and objects with weak characteristics from region 5 to region 8, the contrast methods can hardly perceive the objects, which causes plenty of missing detection. However, the proposed method can achieve the best detection performance for the above complex without missing and false detection.

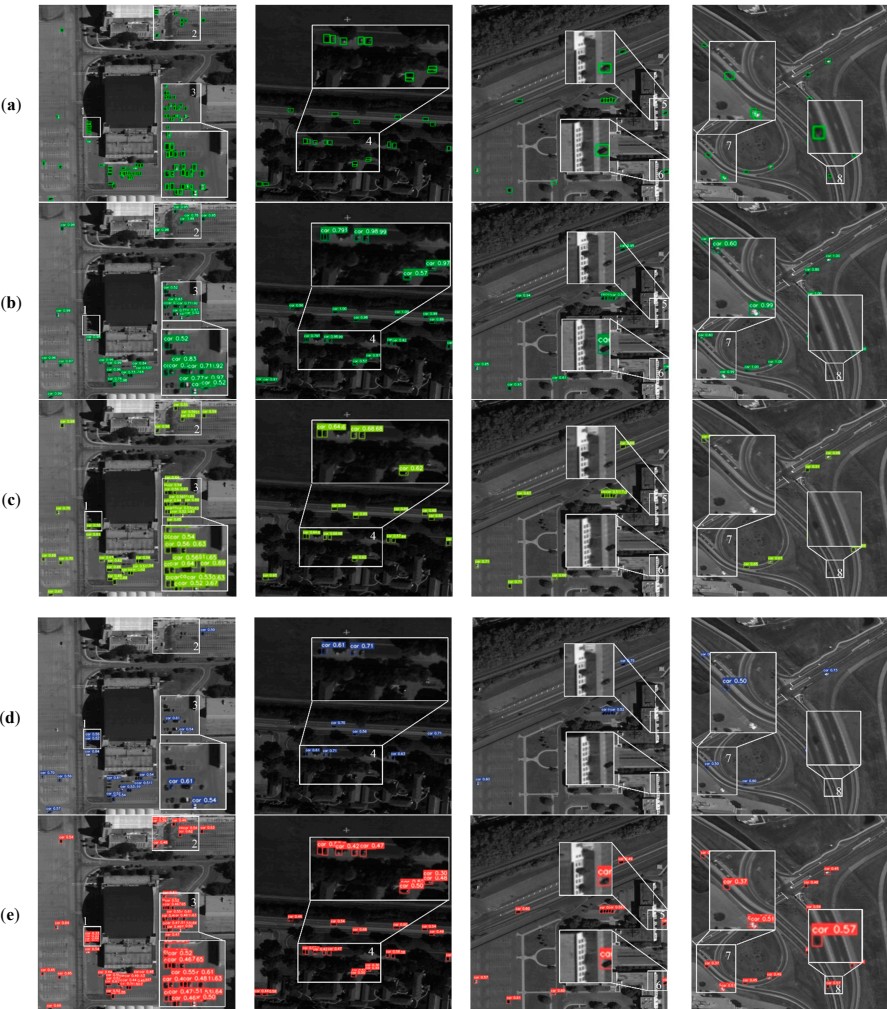

**Figure 9.** Visualization of the detection results for different methods. (**a**) Input images with ground truth, (**b**) Cascade-RCNN, (**c**) imYOLOv3, (**d**)FII-CenterNet, and (**e**) our method.

## 5. Conclusions

In this paper, we proposed an anchor-free detector named FE-CenterNet aiming at small and dim object detection in complex remote sensing scenes. First, we introduced the multiscale contextual information to suppress the interference of false alarms similar to the objects and integrate a coordinate attention mechanism to improve the perception of small objects, thus proposing the FEM. Then, to improve the positioning regression accuracy, we proposed a new loss function that combines the original loss function of CenterNet with CIOU loss. Finally, to verify the detection performance, we constructed the DSVD, composed of varied kinds of complex scenes and objects. The experimental results show that our method has a better detection performance and achieves a higher inference speed than the other typical algorithms, proving its potential for small object detection in complex remote sensing scenes.

**Author Contributions:** Conceptualization, T.S. and J.G.; methodology, T.S. and J.H.; software, T.S.; validation, T.S., P.Z., and G.B.; formal analysis, Y.Z.; investigation, J.G.; writing—original draft preparation, T.S. and J.G.; writing—review and editing, J.H.; supervision, X.Z. and W.Z. All authors have read and agreed to the published version of the manuscript.

**Funding:** This work was supported by the National Natural Science Foundation of China (NSFC) (62101160).

**Data Availability Statement:** The dataset established is available upon requests from the corresponding author.

**Conflicts of Interest:** The authors declare no conflict of interest.

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
