# Peer review of "Feature-Enhanced CenterNet for Small Object Detection in Remote Sensing Images"

_remotesensing, doi:10.3390/rs14215488_

Round 1

Reviewer 1 Report

The manuscript is interesting. The paper can only be accepted if the authors agree to at least make the dataset open-sourced otherwise the results cannot be verified. 

1. Remove the word "novel" from the manuscript; whether something is novel or not is for the scientific community to decide and not by the authors.

2.  The abstract needs a rewrite. The authors state, "Experiment results demonstrate that the proposed method performs better than the mainstream object detectors and provides a speed advantage at the same time." The authors need to provide quantifiable numbers like, how much % better and faster; simply saying something is fast does not mean anything. 

3.  The dataset cannot be "The dataset established is available upon requests from the corresponding author." Sorry, but the dataset needs to be made open-sourced otherwise it challenges the validity of the results. The code accompanying the dataset can be made "available upon request".  Though, in good faith, it is advisable to make both the dataset and the code open-sourced.

4. I see that the data set that the authors show are clear images, I am interested to know how would the results would change for "hazy", "noisy"  and "blurry" image datasets. Please elaborate and provide a discussion. 

Reviewer 2 Report

The objectives are well defined, fitting in the scopes of thejournal.The article is written in an appropriate way andtheconclusions interesting for the readership.

It seems that the direction and research method pursued by the paper are appropriate for the paper.

Reviewer 3 Report

In this paper, you proposed a novel anchor-free detector named FE-CenterNet aiming at small and dim object detection in complex remote sensing scenes. But I still have some questions:

1. In Lines 51, CenterNet [11], as a Typical Representative of Anchor-Free detectors. Why introduce CenterNet specifically, is it good for small object detection? In other words, there are many models that can be applied to small object detection, such as RefineDet, YOLOv5, etc. In my opinion, Introduction does not highlight the current problems with small goals.

2. In Lines 64, Scalar objects and complex scenes We construct the dim and small vehicle dataset (DSVD) composed of various objects and complex scenes. There was no supplemental material, and the experiment was done on a publicly available data set, which I doubt.

3. Chapter 2.1 mainly introduces that object detection algorithms based on deep learning can be distinguished according to two-stage and single-stage detection algorithms, and I have not seen more studies applied to small object detection. As far as I know, many people have achieved good results in small object detection using different networks.

4. Chapter 3.1 describes the network in this paper. Is this the first innovative network for the FE-CenterNet network? Many people build the feature enhancement module FEM to join different networks, and the attached article also improves the network, which is not highly innovative.

5. In Lines 394, for Figure 5 Visualization of the Detection Results In ablation experiments. It was only compared with the CenterNet network, so it was not accurate. And the detection effect of the enlarged white box area is not high.

6. In Lines 426, it is clearly Figure 6. Secondly, for Figure 6. Visualization of the Detection Results for Different Methods. The experimental results compared with the attachment, the effect is not very good.

In general, this paper has improved the algorithm, I do not recommend the article for publication, it needs a major revision.

Reviewer 4 Report

This paper addresses the problem of detection of small objects in remote sensing images. It is based on a deep neural network architecture known as Centernet. Authors introduce what they call Feature Enhancement Module (FEM) to the Centernet architecture. This module extracts contextual information  at multiple scales and  combines them using a mechanism of the style of self-attention.  The paper also proposes a loss function known as CIOU to address the problem of small object detection.

1. Language:  The quality of the language is quite good. Grammatical and style errors are rare and the text is eminently understandable. The paper is  well organized and generally clearly written. Please define the acronyms in their first occurance (eg: CIOU line 289).

2. Introduction: A good introduction is provided.  However, the description of the state-of-the-art is too general. The authors could have cited more work on small object detection in order to position their work with respect to the previous work. 

3. Method: In order to tackle detection of small objects, the authors suggest using more contextual information. This is in effect done by employing dilated convolutions at different scales. This is in a way sounds a bit counter-intuitive, because you look at larger scales to enhance detection of smaller scale objects. This approch has the risk that it learns patterns around the target objects rather than the target objects themselves. This can lead to cases where the system fails to detect objects of interest when the typical context is absent. In the opposite case, the system can report detections if such learned patterns are present even when the target objects are not in the image. One can easily test this by examining the detection performance with the target objects masked.

In lines 309-310,  it is mentioned that in the case of Centernet, the center position and target size are optimized independently. In order to avoid the disadvantages caused by this the authors propose an additional loss function in equation 16. However, in this proposed loss function also the center position and target sizes are independent in the sense that they are incorporated in two different terms. Therefore, it is better if authors can clarify what is meant by independence of center position and target size.           

4. Results: In the ablation study, the authors could have studied the effect of feature aggregation and attention generation seperately. It is interesting know which component of the Feature Extraction Module (FEA) has the most impact.

5. Conclusions: Related to the issue mentioned in point 3 above, it is difficult to know why the system gives an improved performance. The case could be that the system genuinely learns small objects and their contexts, or it focuses too much attention on the context, which is undesirable.

Some specific comments on presentation and format:

(a) In equation (7) define r_x,y

(b) In line 277 is r_encode the same as v_encode?

(c) In Eqn 11  w_y^k(k,j)  not w_y^k(k,y).

(d) Define acronyms (CIOU) 289

(e) Figure 5 (format error one part appears in the previous page. Discussion refering to Figure should contain precise references to the details.

(f) Table 2 format error, Table 2 parts on different pages)

(g) Figure 6 (format error, parts on different pages, wrong figure number , Fig 1 instead of Fig 6)

Reviewer 5 Report

This paper proposed a feature-enhanced CenterNet for small object detection of RS images. There are some flaws for the paper such as:

1.      The proposed FEM (FAS and AGS) and the improvement of the loss function lacks of innovation. The FAS seems like ASPP plus SE, and the author does not clearly cite or describe the method. It’s not clear that the steps and formulations are original or cited.

2.      The author should make melt tests and more figures for the experiment part to prove the superiority of the improved models. The comparisons with other models is relatively less, some recently models such as Yolov5/v6.

3.      More discussion should be added to enable readers to have a more comprehensive understanding of proposed algorithm.

4.      I think “Datasets and Evaluation Metrics” in section 4.1 should be introduced Independently.

Round 2

Reviewer 1 Report

The authors made satisfactory revisions to the manuscript. I am happy to recommend it for publication in the present form. 

Reviewer 3 Report

The article has been modified better this time. Please check carefully for some grammar problems.

Reviewer 5 Report

The author has done improvements in the updated version which solved the queries in the review comments.  Some minor modifications are necessary.

1)      What’s the definition of “small object” in this paper? As for a car, maybe it’s a small object in the 1m resolution satellite image, while it’s a big object observed from ground cameras. There are a series of papers aimed at small object detection. It’s strongly suggested to read their work for reference.

2)      Only one type of object –vehicle, is discussed. The paper title is more suitable for “small vehicle detection” instead of “small object detection”. If the author wants to remain the title, more types of small object must be tested.
